# Interaction of Potato Autophagy-Related StATG8 Family Proteins with Pathogen Effector and WRKY Transcription Factor in the Nucleus

**DOI:** 10.3390/microorganisms13071589

**Published:** 2025-07-05

**Authors:** Sung Un Huh

**Affiliations:** Department of Biological Science, Kunsan National University, Gunsan 54150, Republic of Korea; sungun@kunsan.ac.kr; Tel.: +82-63-469-4587; Fax: +82-63-469-7421

**Keywords:** autophagy-related gene (ATG), *Ralstonia* effector PopP2, WRKY transcription factor, autophagosome

## Abstract

Autophagy is an essential eukaryotic catabolic process through which damaged or superfluous cellular components are degraded and recycled via the formation of double-membrane autophagosomes. In plants, autophagy-related genes (ATGs) are primarily expressed in the cytoplasm and are responsible for orchestrating distinct stages of autophagosome biogenesis. Among these, ATG8 proteins, orthologous to the mammalian LC3 family, are conserved ubiquitin-like modifiers that serve as central hubs in selective autophagy regulation. Although ATG8 proteins are localized in both the cytoplasm and nucleus, their functions within the nucleus remain largely undefined. In the present study, the ATG8-interacting motif (AIM) was identified and functionally characterized in the potato ATG8 homolog (StATG8), demonstrating its capacity for selective target recognition. StATG8 was shown to form both homodimeric and heterodimeric complexes with other ATG8 isoforms, implying a broader regulatory potential within the ATG8 family. Notably, StATG8 was found to interact with the *Ralstonia solanacearum* type III effector PopP2, a nuclear-localized acetyltransferase, suggesting a possible role in effector recognition within the nucleus. In addition, interactions between StATG8 and transcription factors AtWRKY40 and AtWRKY60 were detected in both cytoplasmic autophagosomes and the nuclear compartment. These observations provide novel insights into the noncanonical, nucleus-associated roles of plant ATG8 proteins. The nuclear interactions with pathogen effectors and transcriptional regulators suggest that ATG8 may function beyond autophagic degradation, contributing to the regulation of nuclear signaling and plant immunity. These findings offer a foundational basis for further investigation into the functional diversification of ATG8 in plant cellular compartments.

## 1. Introduction

Autophagy is a phenomenon within an intracellular quality control system [1]. It is a process in which abnormal cellular components, such as protein aggregates and damaged organelles, are enclosed within double-membrane structures called autophagosomes [2]. These autophagosomes then fuse with lysosomes in animals or vacuoles in plants to degrade their enclosed targets. This evolutionarily conserved process has been identified in eukaryotic animals, plants, and fungi [3,4,5]. Studies of a group of evolutionarily similar autophagy-related genes, such as *MAP1LC3/LC3 (microtubule-associated protein 1 light chain 3)-II* in animals, *Autophagy-related gene 8* (*ATG8*) in plants, and *Atg8* in *fungi*, are used as indicators of autophagosome formation [6,7,8,9]. Fungi and insects typically have one *Atg8*, whereas in plants, the presence of 5 to 10 copies of *ATG8* genes suggests significant gene duplication [8,10,11]. This implies that the function of ATG8 may appear in a more expanded form in plants [12].

In plants, for the first time, it was reported that the replication initiation protein C1 of Tomato leaf curl Yunnan virus (TLCYnY) induced plant autophagy. After a direct interaction between the C1 protein of TLCYnY and tomato SlATG8h, C1 translocates from the nucleus to the cytoplasm and undergoes degradation through an autophagy-mediated pathway [13]. Additionally, there have been reports that plant ATG8 interacts with the WRKY transcriptional regulator and might act as a cotranscriptional activator in the nucleus [14,15]. Autophagy-related protein ATG6 has been recognized as a key contributor to plant immune responses. Recent studies have revealed that ATG6 also functions within the nucleus [16]. Specifically, ATG6 has been shown to directly interact with *Nonexpressor of pathogenesis-related genes 1* (*NPR1*), enhancing its nuclear accumulation and protein stability. This interaction promotes the formation of salicylic acid (SA)-induced NPR1 condensates and synergistically upregulates pathogenesis-related gene expression. Functional analyses further demonstrate that the co-overexpression of ATG6 and NPR1 significantly enhances resistance to *Pseudomonas syringae*, highlighting the essential role of ATG6 in modulating NPR1-dependent plant immunity [16]. These findings suggest that autophagy proteins may shuttle between the nucleus and cytoplasm, regulating the stability and degradation of target proteins.

To outcompete the fungus *Fusarium graminearum*, the bacterium *Streptomyces hygroscopicus* secretes rapamycin to inhibit the fungal target of rapamycin (TOR), resulting in the degradation of the histone acetyltransferase Gcn5 through the 26S proteasome. Gcn5 negatively regulates fungal autophagy by acetylating ATG8 and impeding the cellular relocation of ATG8 within the nucleus [17]. This suggests that in eukaryotes, ATG8-related proteins have more diverse functions in the nucleus.

In the previous study, the potato StATG8 family was characterized [15]. Based on this, a phylogenetic analysis was conducted using ATG8 protein sequences from Solanaceae crops and *Arabidopsis thaliana*, revealing that they are grouped into two distinct clades. Potato *StATG8-2.1* and *StATG8-2.2* exhibited differences in their DNA sequences, but their amino acid sequences were completely identical. Although the AtATG8f and AtATG8h formed different clades, a recent study demonstrated that *AtATG8f* and *AtATG8h* are highly responsive under low phosphate (Pi) starvation, and this activation is suppressed in a *phosphate response 1* (*phr1*) mutant [18]. The overexpression of *AtATG8a*, *AtATG8e*, *AtATG8f*, and *AtATG8g* stimulates the formation of autophagosomes and increases autophagic flux [19]. String software (https://string-db.org/) was utilized to explore the interacting network, excluding predictions, with potato StATG8-2 obtained through text-mining and journal searches [20]. The results of this study have been reported previously, and the authors transiently expressed GFP-StATG8-2 in tobacco and identified various interactors via immunoprecipitation in combination with mass spectrometry (IP-MS) analysis [21]. Most interactors are associated with the function of intracellular cytoplasmic regions required for autophagosome formation, and their consequences are evolutionarily related to autophagy. Interestingly, the authors found the newly discovered interactors, including proteins predicted to be localized in various cellular organelles [21]. This is expected to be helpful in research on organelle-mediated autophagy function. In the IP-MS data, StATG8 proteins were captured by IP-MS, enabling the association between ATG8 family members [21].

While the function of ATG8 in plants is clearly associated with degradation processes occurring in the cytoplasm, studies on the protein targets and functions of ATG8 in the nucleus are still lacking. However, the discovery of additional ATG8 interaction motifs and novel binding sites for autophagy adapters and receptors has recently demonstrated an expansion of ATG8 function [8,22]. ATG8 binding proteins utilize a defined ATG8-interacting motif (AIM or LC3 interaction region [LIR]) that contacts a hydrophobic patch on ATG8 known as the LIR/AIM docking site (LDS). Additionally, a ubiquitin-interacting motif (UIM)-like sequence has been discovered for high-affinity binding to the alternative ATG8 interaction site. The identification of new proteins that interact with ATG8 has further advanced the search for its protein targets and cargo proteins [12,23].

This study analyzed the AIM that enables potato StATG8 to access its targets, revealing that the AIM is also present in StATG8. Based on this, homodimeric and heterodimeric interactions within the StATG8 family were observed. Additionally, StATG8 specifically interacted with *Ralstonia* effector PopP2, which possesses acetyltransferase activity in the nucleus. Furthermore, the association of StATG8 and transcription factor AtWRKY40/AtWRKY60 proteins was observed not only in the nucleus but also in cytoplasmic autophagosomes. Although these results do not fully explain the function of ATG8, this study opens up possibilities for understanding the role of the plant ATG8 in the nucleus.

## 2. Materials and Methods

### 2.1. Plant Materials and Arabidopsis Transgenic Plants

Wild-type *Nicotiana. benthamiana* plants and *A. thaliana* Col-0 plants were grown in a growth chamber controlled at 22–25 °C, 45–65% humidity, and a 16/8 h light/dark cycle [24].

### 2.2. BiFC Assay and Agro-Mediated Transient Gene Expression in N. benthamiana

For the mCherry BiFC assay, the BiFC mCherry vector, which was divided into mRYNE (a.a 1–159) and mRYCE (a.a 160–236), was used [25,26]. StATG8 family genes were cloned into mRYNE and mRYCE BiFC vectors using the appropriate restriction enzyme. AtWRKY40 and AtWRKY60 were cloned into the mRYCE BiFC vector. The PopP2 gene was cloned into the mRYNE BiFC vector, and the corresponding primer set used for the cloning is provided in Appendix A. The mRYNE-StATG8s and mRYCE-StATG8s were co-expressed in 3-week-old *N. benthamiana* leaves at OD_600_ = 0.5, and GFP-StATG8 was additionally co-expressed as an autophagy marker. Furthermore, agro-infiltration was performed with the mRYNE-StATG8 and mRYCE-AtWRKY40 or mRYCE-AtWRKY60 combination. The PopP2 gene was cloned into a plasmid vector harboring a C-terminal HF (His_6_-FLAG) tag [27], and the PopP2-mCherry construct [28] was kindly provided by Jonathan Jones (the Sainsbury Laboratory, Norwich, UK). The GFP-AtATG6 construct was generously provided by Yasin Dagdas (Gregor Mendel Institute of Molecular Plant Biology, Wien, Austria).

Agro-mediated transient gene expression was conducted by introducing T-DNA constructs containing the *Agrobacterium tumefaciens* strain GV3101 into 3-week-old *N. benthamiana* plants through syringe infiltration. The agro-cells were combined in a 1:1 ratio at OD_600_ = 0.5 in an infiltration buffer (composed of 10 mM MgCl_2_, 5 mM 2-[n-morpholine]-ethanesulfonic acid [MES], pH 5.6) [29]. Three days after agroinfiltration, the reconstituted mCherry signals from the BiFC assay, along with GFP fluorescence and DAPI nuclear staining, were visualized using confocal laser scanning microscopy. Imaging was performed using a ZEISS LSM 900 (Carl Zeiss Microscopy GmbH, Jena, Germany) and a K1-fluo RT system (Nanoscope Systems Inc., Daejeon, Republic of Korea). For fluorescence detection, the following excitation/emission wavelengths were used: 587/610 nm for mCherry, 488/509 nm for GFP, and 358/461 nm for DAPI.

### 2.3. Yeast Two-Hybrid Assay and Quantitative Liquid β-Gal Assays

The matchmaker GAL4 two-hybrid system 3 (Clontech, Palo Alto, CA, USA) was utilized for yeast two-hybrid interactions. The *StATG8* gene was amplified and cloned into the pGBKT7 vector, incorporating the GAL4 binding domain (BD). The *StATG8s* and *AtWRKYs* were cloned into the pGADT7 activation domain (AD) expression vector. The AH109 yeast strain was cotransformed with the appropriate pGBKT7 and pGADT7 clones and grown in minimal synthetic dropout (SD) medium SD/-Leu/-Trp. To assess interaction strength, cells from SD medium plates were replica-plated sequentially with increasing stringency of selection and analyzed for growth on triple dropout SD/-Leu/-Trp/-His medium.

Quantitative liquid β-gal assays were conducted following the manufacturer’s manual (Clontech, Palo Alto, CA, USA) using o-nitrophenyl-β-d-galactopyranoside (ONPG) as the substrate. The selected cells were inoculated into SD/-Leu/-Trp/-His liquid medium and grown at 28 °C with vigorous shaking until the OD_600_ reached 0.5–1. After pelleting and lysing cells with glass beads, the activity was estimated using the formula 1000 × (OD_420_/t × v × OD_600_), where OD_420_ represents product absorbance, OD_600_ denotes cell density, t signifies the incubation time in minutes, and v is the volume of cells in milliliters.

### 2.4. Co-Immunoprecipitation (Co-IP) and Western Blot

Co-IP and western blot analyses were carried out following standard procedures [27,30]. Proteins were transiently expressed in 3-week-old *N. benthamiana* leaves. Samples were harvested at 2 dpi and ground using a mortar and pestle in liquid nitrogen (LN2). Total proteins were extracted using the extraction buffer [25 mM Tris-HCl, pH 7.5, 150 mM NaCl, 1 mM EDTA, 10% glycerol, 10 mM DTT, 0.2% Nonidet-40, 2% polyvinylpolypyrrolidone and protease inhibitor cocktail (Roche Diagnostics, Mannheim, Germany)]. Samples were centrifuged at 15,000× *g* for 25 min at 4 °C, and co-IP was performed on the supernatant with FLAG beads. For Western blots, samples were boiled for 10 min in 3 × SDS sample loading buffer (25 mM Tris-HCl, pH 6.8, 300 mM DTT, 6% SDS, 0.3% bromophenol blue, and 30% glycerol). Proteins were separated by 10% SDS-PAGE and transferred to PVDF membranes using the Trans-Blot Turbo Transfer System (Bio-Rad Laboratories, Hercules, CA, USA). Immunoblotting was carried out with HRP-conjugated anti-GFP (Santa Cruz Biotechnology, Dallas, TX, USA), anti-acetyl-lysine antibody (Cell Signaling Technology, Danvers, MA, USA), and anti-FLAG (Sigma-Aldrich, St. Louis, MO, USA). The experiments were repeated at least three times with similar results.

### 2.5. Statistical Analyses

The statistical analyses were performed using a one-way ANOVA test in origin 2021b (OriginLab Corporation, Northampton, MA, USA). *** *p* < 0.001 (compared with WT), one-way ANOVA test.

## 3. Results

### 3.1. StATG8 Is Primarily Localized in the Cytoplasm and Nucleus of Primary Roots in Stable GFP-StATG8 Transgenic Plants

In confocal microscopy observations involving plant ATG8, the presence of ATG8 in the nucleus was not considered a significant phenomenon. This is because attention is focused on observing signals in the form of structures such as autophagosomes in the cytoplasmic region. To confirm the presence of StATG8 in the nucleus, GFP-StATG8-2.1 was transiently expressed in *N. benthamiana*. After 3 days, samples were stained with DAPI, and the subcellular localization of StATG8 was observed by confocal microscopy (Figure 1a). StATG8 was observed in punctate structures similar to autophagosomes. Additionally, GFP-StATG8 protein also has DAPI staining at the same location. However, in most autophagy observations, the nucleus is often not visible. To confirm this, Z-stacking was performed to create a 3d view of the cell space. The resulting spatial differences between the location of autophagosomes and the nucleus within the cell can be easily observed (Figure 1b).

StATG8 family genes fused with GFP at the N-terminus were used to generate transgenic plants in *A. thaliana*. The GFP signals in the primary roots of the stable *GFP-StATG8* T3 plants, in the absence of stress or stimulation, were observed in the cytoplasm and nucleus patterns (Figure 2). In the *GFP-StATG8-2.1* transgenic plant, the intracellular distribution of GFP at the root tip, where various signaling events such as development occur, is primarily observed in the cytoplasm and autophagosomes (Appendix A). StATG8-2.1 proteins were also detected in the cytoplasmic–nucleus regions (Appendix A). The StATG8 protein in the nucleus could move to the cytoplasm through a shuttling event (Appendix A). These results suggest that ATG8 might undergo movement between the nucleus and cytoplasm through post-translational regulation, such as acetylation or deacetylation, as observed in animal models [31].

### 3.2. StATG8 Forms Homodimers

Based on the reported capture data of StATG8 proteins from the IP-MS [21], putative canonical AIMs are analyzed in StATG8 proteins via the iLIR autophagy database [32]. Additionally, the two-dimensional sequences of StATG8 protein are analyzed using ESPrip [33]. In Appendix A, human lc3a (q9h492) had only one putative AIM in the β-3 region. In contrast, the AIM of the StATG8 protein was predicted to be identical in β-3, a position similar to the putative AIM of LC3. Additional AIMs were also predicted, overlapping AIMs in the α-5 region of StATG8-1.1, StATG8-2.1, StATG8-3.1, and StATG8-3.2 proteins (Appendix A). Based on this analysis, the Y2H experiment was performed. A combination of AD-StATG8-1/BD-StATG8-1.1 or AD-StATG8-2.1/BD-StATG8-2.1 or AD-StATG8-3.1/BD-StATG8-3 or AD-StATG8-4/BD-StATG8-4 showed significantly higher β-gal activity compared to the control group (Figure 3a). Thus, StATG8, which has its own putative AIMs, may exist as a homodimer.

To confirm the Y2H experiment, a BiFC assay was performed. In all cases, split mCherry was fused to StATG8 at the N-terminal, and mRYNE-StATG8 and mRYCE-StATG8 were constructed. Since ATG8 is an autophagy marker, control GFP-StATG8 was co-expressed, respectively. As shown in Figure 3b, reconstituted mCherry red signals were observed through a confocal microscope in all combinations and merged with the GFP signal of GFP-StATG8, as expected. Additionally, the negative control did not produce any nonspecific BiFC signal, supporting the specificity of the observed protein–protein interactions (Appendix A). Thus, StATG8 forms homodimers in the cytoplasm, nucleus, and autophagosome.

### 3.3. StATG8 Forms Heterodimers

Previous results confirm that StATG8 homodimerization is possible, as at least two to three AIMs exist in StATG8, although the actual active AIM has not yet been determined. This possibility could lead to the formation of heterodimers among the StATG8 family. The BiFC experiments revealed that mRYNE-StATG8-1.1/mRYCE-StATG8-2.1 or mRYNE-StATG8-1.1/mRYCE-StATG8-3.1 or mRYNE-StATG8-1.1/mRYCE-StATG8-4 combinations were transiently co-expressed with control GFP-StATG8 in *N. benthamiana*. The mRYNE-StATG8-1.1 showed reconstructed mCherry BiFC signals along with mRYCE-StATG8-2.1, mRYCE-StATG8-3.1, and mRYCE-StATG8-4 (Figure 4). Likewise, mRYNE-StATG8-2.1 showed BiFC signals with mRYCE-StATG8-3.1 and mRYCE-StATG8-4. The co-expression of mRYNE-StATG8-3.1 and mRYCE-StATG8-4 resulted in the observation of mCherry BiFC (Figure 4). Thus, StATG8 forms heterodimers and exists in the nucleus, cytoplasm, and autophagosome.

### 3.4. The Ralstonia Effector PopP2 Directly Associates with StATG8 and Relocalizes to Autophagosomes

Among the effector proteins possessed by *Ralstonia*, PopP2 is the protein whose function is best studied [30,34,35,36]. PopP2 is an enzyme with acetylation activity and is known to suppress plant immunity by weakening WRKY DNA binding activity through the acetylation of specific lysine residues on plant WRKY transcriptional regulators and resistance protein in the nucleus [27,37]. Additionally, the *Arabidopsis* cysteine protease RD19 has been reported to relocalize to the nucleus together with PopP2 in a mobile vacuole-related compartment [38].

Is PopP2 actually located in the nucleus? When the *35S::PopP2-mCherry* construct was expressed in *N. benthamiana*, predominantly nuclear subcellular localization was observed (Figure 5a). However, PopP2 was also detected in the cytoplasm of some cells (Figure 5a and Appendix A). This phenomenon may be an artifact due to the use of a strong promoter, but the 35S promoter was utilized in most functional studies on PopP2. In this study, PopP2 mainly functions in the nucleus and may also exist, to some extent, in the cytoplasm.

In a previous study, it was reported that StATG8 interacts with the PopP2 protein [39]. To investigate alterations in subcellular localization, GFP-StATG8-2.1 and PopP2-mCherry were transiently co-expressed in *N. benthamiana*, and their co-localization was confirmed using confocal microscopy. The Figure 5b images show that StATG8-2.1 and PopP2 mainly co-exist in the nucleus, and the subcellular localization of PopP2 was also observed in some cytoplasmic regions. Additionally, co-localization was observed in specific autophagic body types. To further confirm, BiFC was employed. mRYNE-PopP2 and mRYCE-StATG8-2.1 were co-delivered, and GFP-StATG8-2.1 was additionally co-expressed as an autophagy marker. In Figure 5c, reconstructed mCherry BiFC signals were observed in the cytoplasm, nucleus, and autophagic bodies. Similarly, in the BiFC of the mRYNE-PopP2 and mRYCE-StATG8-3.1 combination, BiFC signals were found in the cytoplasm, nucleus, and autophagic bodies (Figure 5c and Appendix A). To reconfirm this co-localization and BiFC, co-IP was performed. *Arabidopsis* GFP-AtATG6 and GFP empty vector were used as negative controls. PopP2-HF and GFP-AtATG6 or GFP-StATG8-2.1 or GFP control were subjected to agro-mediated transient co-expression. Three days later, samples were harvested and subjected to immunoprecipitation with FLAG beads. In Figure 5d, it is shown that only StATG8-2.1 specifically interacts with PopP2, but not ATG6. Additionally, upon verifying the specificity of acetylation activity towards StATG8, it was observed that PopP2 acetylated only StATG8-2.1 and not AtATG6 (Figure 5e). Therefore, it can be concluded that StATG8 specifically interacts with PopP2 in the nucleus, cytoplasm, and autophagosomes.

### 3.5. StATG8 Interacts with the WRKY Transcription Factor

Previous studies have shown that WRKY, a transcriptional regulator, interacts with ATG8, and it has recently been reported that human lc3b interacts with the transcription factor LMX1B and acts as a transcriptional cofactor [40,41]. Based on this, putative candidates with AIM among *Arabidopsis* transcription factors were analyzed in the iLIR autophagy database. As a result, about 100 transcription factors contained valid putative AIMs (Appendix A). Accordingly, in a previous study, AtWRKY40 and AtWRKY60, which were co-IPed with StATG8-2.1, were selected [15]. AtWRKY40 has AIMs in the N-terminal and C-terminal regions. AtWRKY60 included one AIM (Figure 6a).

Based on this, the protein interaction of StATG8-AtWRKY40 and StATG8-AtWRKY60 was confirmed through y2h. Yeast cells selected by co-expressing BD-StATG8-1.1/AD-AtWRKY40 or BD-StATG8-1.1/AD-AtWRKY60 showed significantly stronger liquid β-gal activity than the control BD-StATG8-1.1/AD-empty combination (Figure 6b). Similarly, Y2H assays were performed for StATG8-2.1 with AtWRKY40 and AtWRKY60, both of which had previously been tested by co-IP [18], was also performed. In cells co-expressing the BD-StATG8-2.1/AD-AtWRKY40 or BD-StATG8-2.1/AD-AtWRKY60 combination, stronger β-gal activity was observed than in the control StATG8-2.1/AD-empty vector (Figure 6b). As a result, StATG8 interacts with WRKY present in the nucleus.

BiFC was performed to further study the subcellular location. StATG8 was constructed with split mCherry (mRYNE-StATG8) at the N-terminal, and AtWRKY was fused with split mCherry (mRYCE-AtWRKY) at the N-terminal. Whether mRYNE-StATG8-1.1/mRYCE-AtWRKY40 and the autophagy marker GFP-StATG8-1.1 were co-expressed, interestingly, it was observed in the cytosol, nucleus, and a few autophagic bodies (Figure 6c). Likewise, when the mRYNE-StATG8-1.1/mRYCE-AtWRKY60 combination was also expressed in *N. benthamiana*, it was observed in the cytosol, nucleus, and a few autophagic bodies (Figure 6c). Thus, StATG8 can bring WRKY from the nucleus into the cytoplasm. StATG8-2.1 also performed BiFC analysis using AtWRKY40 or AtWRKY60. Reconstituted mCherry BiFC signals were observed in the nucleus, cytoplasm, and autophagosomes of mRYNE-StATG8-2.1/mRYCE-AtWRKY40 and mRYNE-StATG8-2.1/mRYCE-AtWRKY60 (Figure 6d). Additionally, no nonspecific BiFC signal was observed in the negative control (Appendix A). StATG8 has the capability to translocate WRKY proteins from the nucleus to the cytoplasm. This would imply a regulatory role for StATG8 in the localization of WRKY proteins, potentially affecting their function or activity. This observation could have implications for understanding the interplay between autophagy-related processes and transcriptional regulation mediated by WRKY proteins.

If StATG8 transports WRKY out of the nucleus, the formation of autophagosomes should increase. Three days after transiently expressing mRYNE-StATG8-1.1/mRYCE-AtWRKY40 or mRYNE-StATG8-2.1/mRYCE-AtWRKY40 in *N. benthamiana* leaves, the plants were placed in a 150 mM salt solution for 24 h to observe autophagosomes. Merged signals of GFP-StATG8-1.1 and BiFC of mRYNE-StATG8-1.1/mRYCE-AtWRKY40 were observed in the cytoplasm and multiple autophagic bodies (Figure 6e).

StATG8 undergoes homo-/heteromeric interactions within the family. It also showed effects on changes in the subcellular localization of effector PopP2 and WRKY proteins. These results suggest a mechanism by which StATG8 translocates from the nucleus to the cytoplasm through protein interactions to promote protein clearance and stability through autophagy-mediated regulations. It is also possible that ATG8 acts as a cofactor to regulate genes together with transcriptional regulators such as WRKY (Figure 7).

## 4. Discussion

Autophagy contributes to maintaining cellular homeostasis by delivering cytoplasmic cargo to lysosomes (in animals) or vacuoles (in plants) for decomposition and the recycling of damaged proteins or organelles [42]. In macroautophagy (referred to herein as “autophagy”), the sequential action of conserved autophagy-related gene (ATG) products results in the formation of autophagosomes, new specialized organelles that can be produced selectively or non-selectively [43]. The function of autophagy, which mainly occurs in the cytoplasm, is still largely unknown, as is its function in the nucleus. Therefore, among autophagy proteins, understanding how the autophagic response is coordinated in the nucleus and cytoplasm remains a key goal.

Macroautophagy/autophagy-related proteins, LC3/ATG8/Atg8, are evolutionarily conserved proteins crucial for autophagosome biogenesis and essential for the selective degradation of various substrates [44]. Recently, the nuclear functions of ATG8, an autophagy-related protein, have been discovered in animals, fungi, and plants [13,17,40]. Since the phenomenon called autophagy occurs in the cytoplasm, the simplified model suggesting that the ATG8 protein shuttles between the nucleus and cytoplasm through acetylation in the nucleus was proposed to expand the understanding of ATG8 functions. However, it is quite interesting that nuclear-related proteins have not yet been detected in screenings such as liquid chromatography–mass spectrometry (LC-MS), yeast two-hybrid (Y2H), and BirA* tag proximity-dependent biotin identification (BioID) techniques, despite numerous reports of protein interactions with ATG8 [21,45,46].

One certainty is that the ATG8 protein undergoes acetylation/deacetylation by host proteins, facilitating its shuttle between the nucleus and cytoplasm through this post-translational step. Upon examining the potato StATG8 LC-MS screening data conducted by Yasin’s group [21], only histone deacetylase 5, a nuclear protein, was detected. While further study is necessary, this finding suggests that plant ATG8 might be influenced by its transfer from the nucleus to the cytoplasm through histone deacetylation. Similarly, the degradation of the acetyltransferase Gcn5 from the fungus *F. graminearum* is promoted through the 26S proteasome by rapamycin secreted by the bacterium *S. hygroscopicus*. Consequently, the rapamycin-induced degradation of Gcn5 reduces fungal ATG8 acetylation and affects nuclear–cytoplasmic transport, thereby disrupting fungal autophagy homeostasis and inhibiting fungal growth [17]. In this study, the interaction between PopP2, a *Ralstonia* effector with acetyltransferase activity, and StATG8, along with specific acetylation by PopP2, was revealed [39] (Figure 5). Notably, the presence of PopP2 in the autophagic body, coupled with changes in its subcellular localization, strongly suggests degradation through autophagy. This observation raises the intriguing possibility of the co-evolutionary arms race between plants and pathogens [39,47]. It implies that ATG8, initially localized in the cell nucleus, not only plays a role in recognizing and eliminating invading pathogen proteins but is also a target for pathogens, potentially disrupting autophagy homeostasis.

Although it has been reported that the mRNA level of the *ATG* gene is regulated by transcription factors (TFs) [48,49,50,51], studies on the interaction between TFs and ATG proteins in plants are limited [14,15]. Recent studies in animals have revealed the involvement of LC3 as a coactivator of TFs [40]. This TF-ATG8 protein interaction may enable two regulatory mechanisms, including autophagy-mediated TF degradation and coactivation. Although the molecular mechanism is not fully elucidated in this study, the observation of WRKY40-StATG8 complexes in cytosolic autophagosomes suggests autophagy-mediated TF degradation. It is likely that TF degradation and coactivation by autophagy are intertwined, and further studies are underway to investigate this in more detail.

Recent studies highlight the existence of multiple copies of plant ATG8 [15,52]. It is expected to act as a mechanism to maintain intracellular homeostasis through autophagy, protecting against environmental challenges and invaders, unlike its role in animals. Moreover, in addition to the well-established AIM, the discovery of a variety of novel motifs suggests that ATG8 not only plays a unique role in autophagy but may also have additional functions promoted by protein interactions [8]. The ATG8–target protein interaction test, conducted through protein–protein simulation programs such as AlphaFold, can serve as a valuable technique [22].

It is difficult to directly detect LC3/ATG8/Atg8 family members that interact with target proteins in the nucleus. Nevertheless, autophagy may play a role in removing or regulating nuclear-derived proteins. It was reported that ATG8-phosphatidylethanolamine (PE) exists in a lipidized form in a multimeric state [53]. Although the main functions of ATG8 include interaction with cargo target proteins, more complex forms, such as homo/heterodimer formation within autophagosomes, were observed in this study. This complexity suggests that the structural states of ATG8, including homodimeric and heterodimeric configurations, may contribute to functional roles beyond established interactions with target proteins. Despite the overlap in protein interaction targets between LC3A, LC3B, LC3C, and human homologues, these proteins are specific in their targets [54,55,56]. A similar pattern was observed in the potato StATG8 protein expressed in tobacco, showing the existence of target proteins specifically bound between overlapping targets and StATG8 members [21]. This represents functional improvement and evolution despite minimal differences in protein structure or amino acid sequence. The potential occurrence of protein interactions between ATG8 members introduces an additional layer of complexity, requiring further research to ascertain the feasibility of functional expansion attributed to ATG8 homo/heterodimers. This complexity underscores the need for a comprehensive understanding of the intricate mechanisms governing ATG8 functions in various organisms.

While research into nuclear-autophagy mechanisms is ongoing, the exploration of ATG8 function in the cell nucleus is in its infancy. Although initially proposed as a nuclear–cytoplasmic shuttle responsible for removing proteins from the nucleus, recent reports have suggested a broader functional repertoire [13,31]. ATG8 has been identified as a transcriptional cofactor that participates in the regulation of gene expression together with transcriptional regulators [40,41]. In plants, the interaction between MeWRKY and MeATG8 has been shown to activate the transcription of *Pathogenesis-related* (*PR*) genes. Additionally, the overexpression of StATG8 has been reported to confer partial resistance to *Ralstonia* infection [14]. ATG8 may also act as a capture protein, contributing to the clearance of pathogenic proteins [13]. The expanded roles of ATG8 in diverse cellular processes highlight the complexity of ATG8 in nuclear and cytoplasmic contexts.

## 5. Conclusions

This study demonstrated the interaction between StATG8 and PopP2, as well as WRKY, within the nucleus, confirming their presence in the nucleus and autophagic bodies. This marks the initiation of exploring the function of ATG8 in the nucleus, emphasizing the need for more detailed functional studies.

## Figures and Tables

**Figure 1 microorganisms-13-01589-f001:**
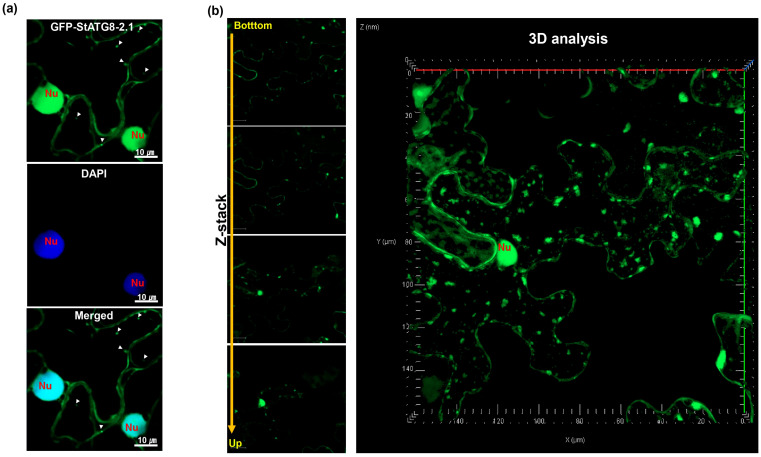
Transiently expressed potato GFP-StATG8-2 protein is found in the nucleus, cytoplasm, and autophagosomes. (**a**). The GFP-StATG8-2.1 construct was introduced into *N. benthamiana* using the agro-mediated transient expression system. Intracellular nuclei were visualized using DAPI staining. Autophagic bodies are indicated by arrows. The nucleus is indicated as ‘Nu’. (**b**). After transient expression of GFP-StATG8-2, the autophagosome region and the nucleus region are distinguishable during confocal observation. Differences in this space were analyzed using Z-stack analysis (ZEN 2.3). The white triangle represents the autophagic body, while the nucleus is indicated as ‘Nu’.

**Figure 2 microorganisms-13-01589-f002:**
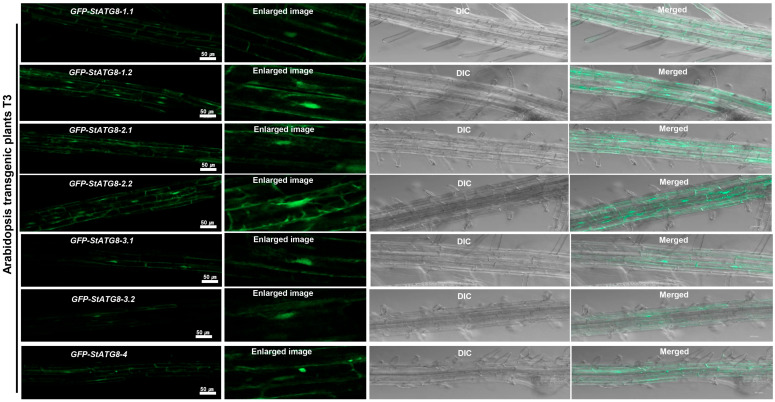
Subcellular localization of the *Arabidopsis* transgenic potato StATG8 line in lateral roots. Seven potato StATG8 family genes were inserted into the 35S::GFP vector and subsequently transformed into *Arabidopsis*. GFP signals in the roots of the *Arabidopsis* seedling T3 transgenic line were observed using a confocal microscope.

**Figure 3 microorganisms-13-01589-f003:**
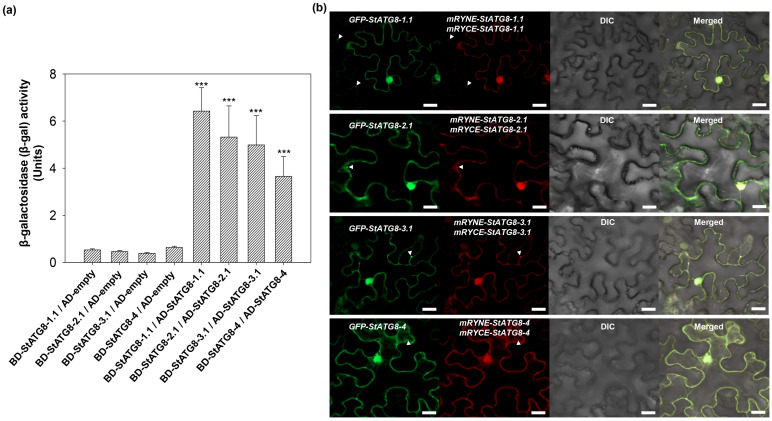
The StATG8 family proteins exhibit homodimeric interactions. (**a**) Quantification of β-galactosidase (β-gal) activity in AH109 yeast cells expressing different binding domain (BD) and activation domain (AD) fusions. The β-gal activity was assessed in AH109 yeast cells expressing various combinations of GAL4 BD and AD fusions. Different plasmid combinations were used for co-transformation. The quantification of β-gal activity was performed in triplicate assays, and the results are presented as average units. The units represent the measured β-gal activity. Data are presented as mean ± SD of three independent samples. *** *p* < 0.001, indicating a statistically significant difference. (**b**) The close proximity of StATG8 family proteins within the cell is confirmed through split-mCherry BiFC analysis, indicating protein–protein interactions. StATG8 was linked to split mCherry (mRYNE or mRYCE) at the N-terminal through cloning. Subsequently, 35S::GFP-StATG8, 35S::mRYNE-StATG8, and 35S::mRYCE-StATG8 constructs were co-expressed in tobacco leaves using agro-mediated transient assays. Three days later, GFP and reconstituted mCherry signals were observed using a confocal microscope. GFP-StATG8 served as an autophagy marker control for intracellular localization. The white triangle represents the autophagic body. A white scale bar indicating 20 μm is shown in the image.

**Figure 4 microorganisms-13-01589-f004:**
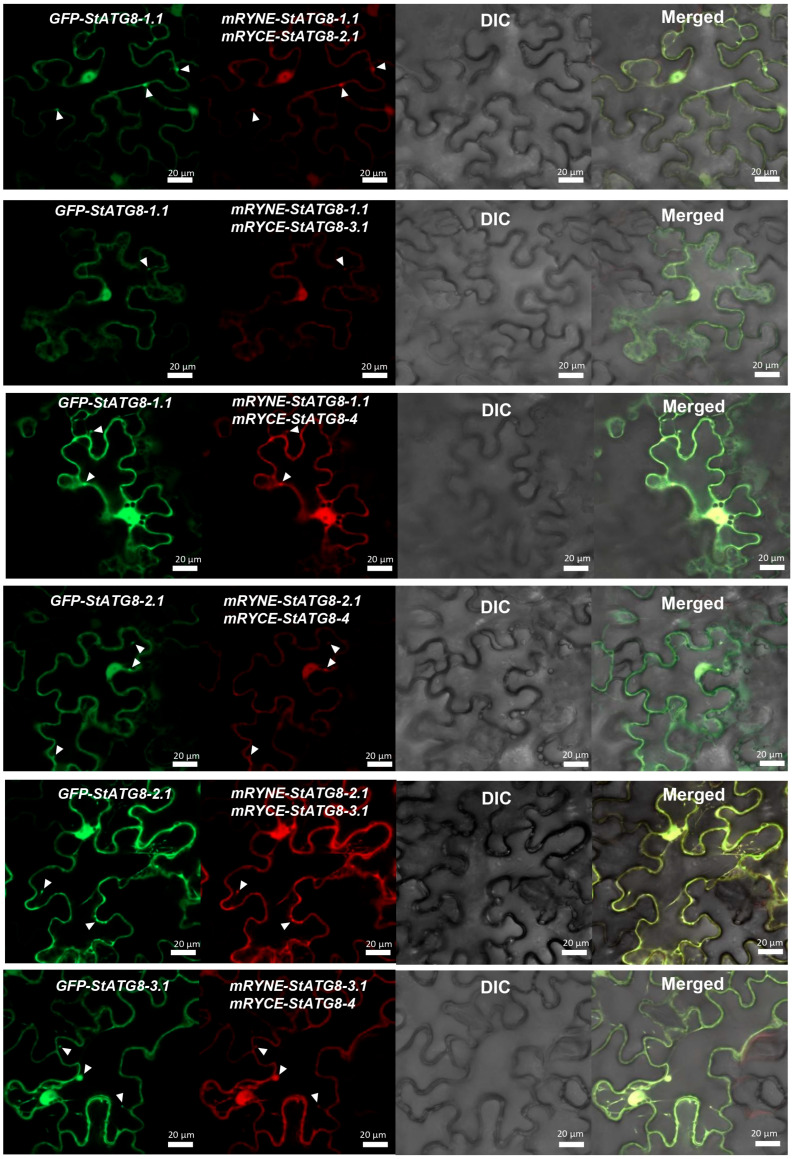
The StATG8 family proteins participate in heterodimeric interactions. All possible combinations of the StATG8 family were systematically generated and introduced into tobacco leaves via transient agro-expression. These combinations included co-expressions such as 35S-mRYNE-StATG8-1.1 with 35S-mRYCE-StATG8-2.1, 35S-mRYNE-StATG8-1.1 with 35S-mRYCE-StATG8-3.1, 35S-mRYNE-StATG8-1.1 with 35S-mRYCE-StATG8-4, 35S-mRYNE-StATG8-2.1 with 35S-mRYCE-StATG8-4, 35S-mRYNE-StATG8-2.1 with 35S-mRYCE-StATG8-3.1, and 35S-mRYNE-StATG8-3.1 with 35S-mRYCE-StATG8-4. Furthermore, each 35S-GFP-StATG8 was co-expressed with all these combinations, serving as a control in the experiment. Three days post-agroinfiltration, GFP and reconstituted mCherry BiFC signals were visualized using a confocal microscope. The white triangle represents the autophagic body.

**Figure 5 microorganisms-13-01589-f005:**
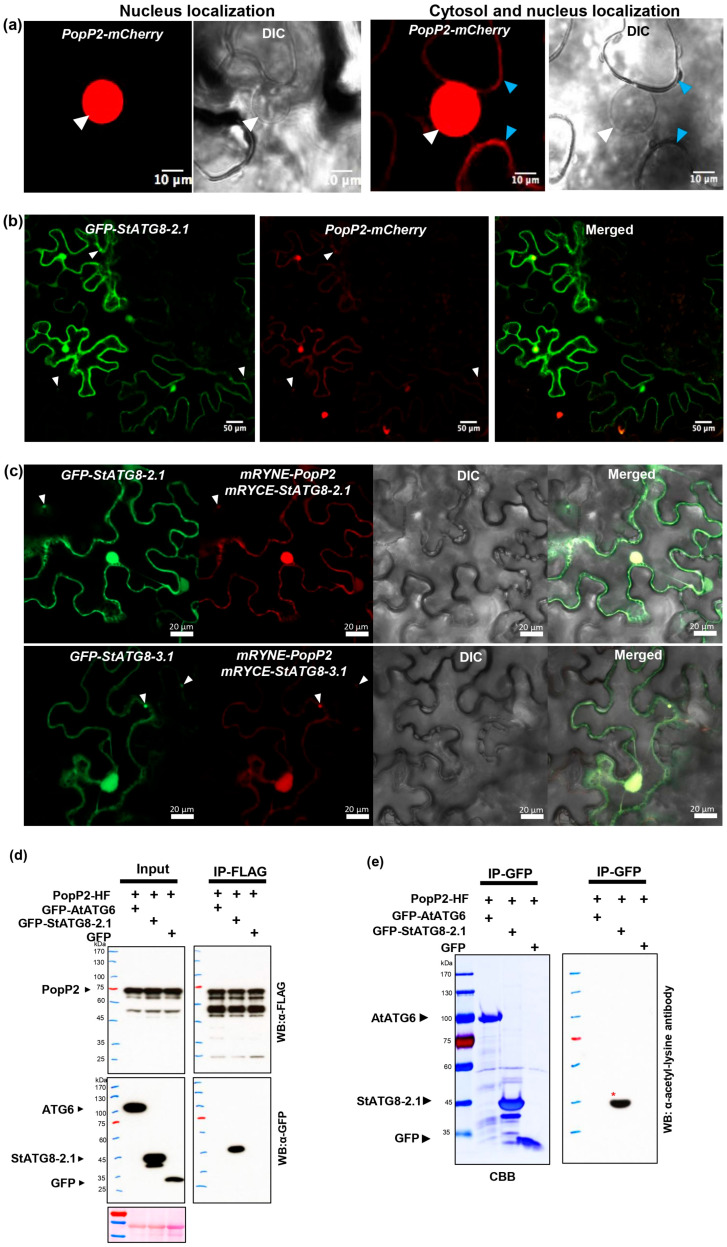
The *Ralstonia* effector PopP2 acetylates StATG8 and relocalizes it to autophagosomes. (**a**) The 35S::PopP2-mCherry construct was introduced into agro-cells, infiltrated into tobacco leaves, and confirmed using a confocal microscope three days later. The white triangle indicates the nucleus, while the blue triangle represents the cytoplasm. (**b**) Co-localization analysis of 35S::PopP2-mCherry and 35S::GFP-StATG8-2.1 in tobacco leaves was performed. Both structures were co-infiltrated into tobacco leaves and observed using confocal microscopy after 3 days. The autophagosome is marked by a white triangle. (**c**) BiFC analysis of PopP2 and StATG8 proteins was conducted. mRYNE-PopP2, mRYCE-StATG8-2.1, and GFP-StATG8-2.1 were co-expressed through agro-mediated transient expression. The white triangle is indicative of the autophagosome. (**d**) Co-IP (co-immunoprecipitation) analysis was performed to investigate the specific association between PopP2 and StATG8. PopP2-HF was introduced via agro-infiltration along with GFP-AtATG6 or GFP-StATG8-2.1 or GFP negative control. After two days, each sample was harvested and subjected to co-IP with FLAG beads. Western blot analysis was conducted using FLAG and GFP antibodies. (**e**) PopP2 specifically acetylates only StATG8-2.1. PopP2-HF was co-expressed with GFP-AtATG6, GFP-StATG8-2.1, or GFP negative control. After two days, total protein was extracted, immunoprecipitated with GFP beads, and subjected to Western blot analysis using an acetylation-specific antibody. Red asterisks indicate acetylated proteins.

**Figure 6 microorganisms-13-01589-f006:**
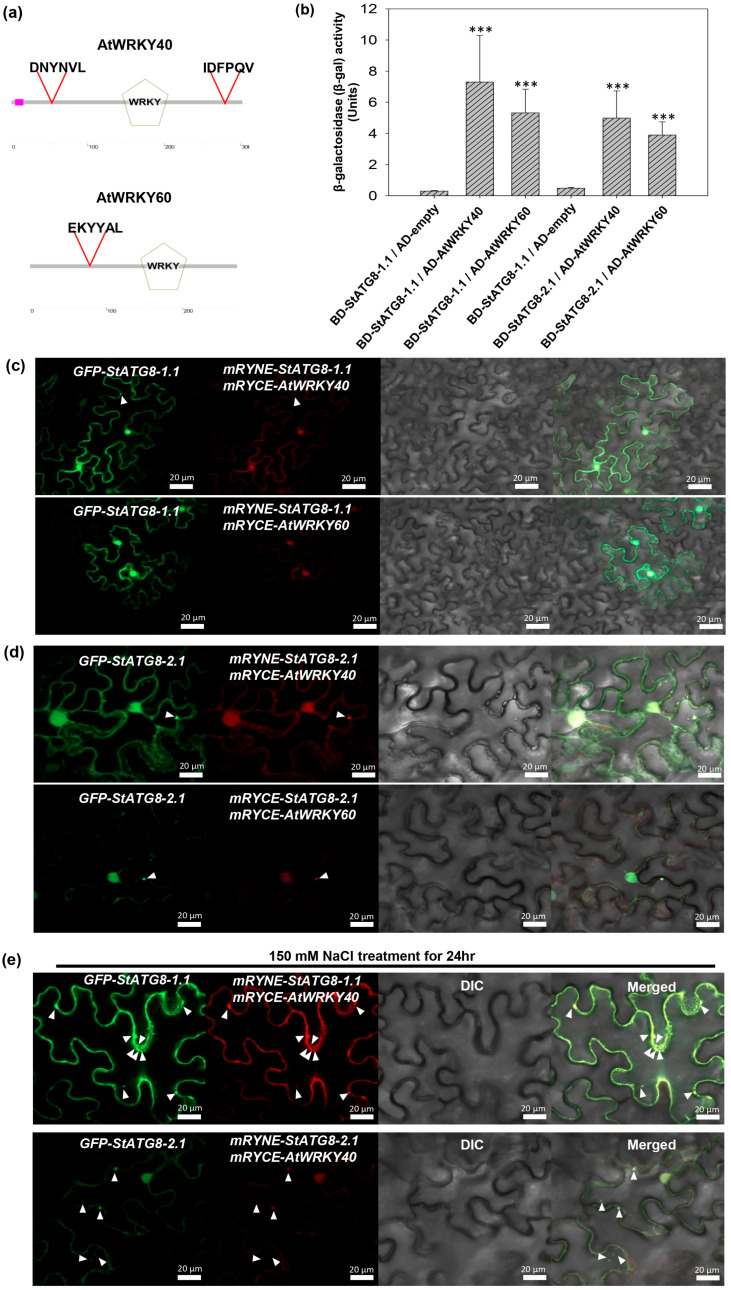
The *Arabidopsis* WRKY protein containing the AIM exhibits close proximity to StATG8 in BiFC analysis. (**a**) WRKY60 and WRKY40 contain a putative ATG8-interacting motif. The iLIR Autophagy Database (https://ilir.warwick.ac.uk/) was employed to in silico identify AIM interaction domain motifs within AtWRKY60 and AtWRKY40. Each AIM is indicated above a red triangle. (**b**) β-gal activity was quantified in AH109 yeast cells co-transformed with various plasmid combinations. Assays were performed in triplicate, and results are shown as mean ± SD of three independent replicates, with activity expressed in enzymatic units. *** *p* < 0.001, indicating a statistically significant difference. (**c**) StATG8-1.1, fused with split NE-mCherry at the N-terminal (mRYNE), AtWRKY40 and AtWRKY60, fused with split CE-mCherry at the N-terminal (mRYCE), were co-expressed in tobacco leaves through agro-infiltration. The mRYNE-StATG8-1.1 co-expressed with mRYCE-AtWRKY40, or mRYCE-AtWRKY60. As a control, GFP-StATG8-1.1 was also co-expressed. White triangles indicate autophagosomes. The reconstructed mCherry signal in red indicates the close proximity of the two proteins. (**d**) In *N. benthamiana* leaf, mRYNE-StATG8-2.1 co-expressed with mRYCE-AtWRKY40, or mRYCE-AtWRKY60 via agro-mediated transient expression. As a control, GFP-StATG8-2.1 was also co-expressed. White triangles indicate autophagosomes. The reconstructed mCherry signal in red indicates the close proximity of the two proteins. The images shown were all taken at 3 days post-agroinfiltration using a confocal microscope. The white triangle represents the autophagosome. (**e**) In *N. benthamiana*, mRYNE-StATG8-1.1 and mRYCE-AtWRKY40 were co-expressed, with GFP-StATG8-1.1 serving as an autophagy marker control. After three days, the plants were irrigated with 150 mM NaCl solution for 24 h. Subsequently, reconstructed mCherry and GFP signals were observed using a confocal microscope. White triangles indicate autophagosomes. The reconstructed mCherry BiFC signal appears in red, and the autophagosome is marked by a white triangle. The datasets displayed in (**c**–**e**) were typical of three independent experiments.

**Figure 7 microorganisms-13-01589-f007:**
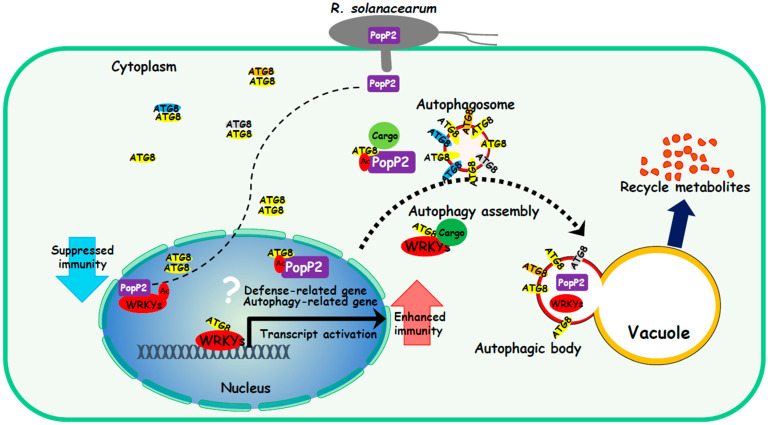
The proposed functional model of ATG8 suggests that it plays an important role in capturing key target proteins, potentially facilitating their sequestration or trafficking for degradation via autophagy. A model is proposed for the function of ATG8 in the nucleus, where it forms a homo-/heterodimeric complex present in both the cytoplasm and nucleus. WRKY proteins, including WRKY40, serve as regulators of plant immunity, with potential positive or negative effects. PopP2 can interact with the WRKY40 protein to suppress plant immunity. ATG8 has the ability to capture WRKY40 in the nucleus and facilitate its clearance or protein stability through cytoplasmic autophagosomes. Similarly, ATG8 exhibits a specific interaction with the effector PopP2 in the nucleus, allowing it to eliminate PopP2 through cytoplasmic autophagosomes.

## Data Availability

The datasets used and/or analyzed during the current study are available from the corresponding author on reasonable request. The iLIR autophagy database was used in this study and can be accessed by anyone. *N. benthamiana* seeds were generously provided by Kyung-Hee Paek of Korea University, and *Arabidopsis thaliana* columbia (Col-0; NASC N1092) background seeds were purchased from the Nottingham Arabidopsis Stock Centre.

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
