# Peer review of "Interaction of Potato Autophagy-Related StATG8 Family Proteins with Pathogen Effector and WRKY Transcription Factor in the Nucleus"

_microorganisms, 2025, doi:10.3390/microorganisms13071589_

Round 1
Reviewer 1 Report
Comments and Suggestions for Authors
The manuscript “Protein interaction of potato autophagy-related StATG8 family proteins with pathogen effector and WRKY transcription factor in the nucleus” promises to reveal some novel facts about Solanum tuberosum ATG8 proteins functioning. It contains interesting results and conclusions, however, several important controls are lacking.
1.Abstract should be re-written and made more accurate. It should contain an introduction sentence on PopP2 effector at least.
- Introduction should be improved. The first paragraph looks separated from the other text, there is no some logical transition from it to the story of autophagy. As to the autophagy part, in my opinion, no need to compare animal and plant cells. Moreover, this part of the introduction doesn’t properly introduce the subject and the object and doesn’t help to understand the problem regarded in the paper. The Intro section lacks clarity and structure as well as some essential information on ATG8 functioning, AIM significance and ATG8 interactions with pathogens effectors. I would recommend to the author to focus on plant autophagy not going into animal and yeast autophagy in details. In the previous paper on potato ATG8 family the introduction section was much more informative and succinct.
- I'm confused with the section 2.1. (Results) and don't understand its purpose in the Results. It looks out of place here as it discusses the previously obtained results on StATG8 and AtATG8 proteins (or it is written in a way that creates such an impression). Thus, it's better to move it to the Introduction.
L 124-126 contains a rather speculative statement on the mechanism underlying shuttling ability of ATG8
- I do not agree with the author that BiFC system could be used to confirm homodimerization. The described experimental setup that includes the same protein fused with split fluorophore is usually used to confirm that the system works properly and both parts of the fluorophore are produced. To make the result a proper basis for the conclusion made by the author an unambiguous negative control should be added to this experiment – for example some mutant variant of StATG8 that lacks a domain essential for interaction with partners. Such a control must be added to that experiment (Fig 3).
- Which of StATG8 was used to generate GFP-StATG8 fuse used for experiments described in sections 2.3, 2.4 and for stably transformed A. thaliana plants?
- I wonder if there is some more accurate marker for autophagosomes, because in all figures where the author use GFP-StATG8 as a marker, the intracellular distribution is not very specific – GFP fluorescence is detected in the nucleus, cytoplasm and (not obvious for me and not convincing) autophagosomes. In addition, it is recommended to apply concavalin A to stabilize autophagosomes (Qi et al., 2023 doi 10.1007/s44307-023-00002-8)
- PopP2-mCherry fuse could produce cytoplasmic fluorescence because of the fusion degradation. To confirm its integrity, the western-blot analysis should be performed that could show if there is some free mCherry or other degradation products present.
The same concern is connected with nucleocytoplasmic distribution of WRKY (Fig 6). First, there should be a control presenting WRKY fused to either part of split fluorescent protein and co-expressed: this combination is expected to give only nuclear fluorescence. Only in this case the conclusion about WRKY re-localization could be made. Second, western-blot analysis of integrity of protein fusions should be confirmed.
- L 266-267 It is written “…putative candidates with AIM among Arabidopsis transcription factors were analyzed…”. However, the paper by Song et al, 2022 claims that StATG8-WRKY interaction is independent of the ATG8 interacting motif (at least in the abstract). Thus the questions arise: 1) why the putative candidate transcription factors were selected based on the presence of AIM? 2) why the search was performed again as it had been already described in Song et al., 2022?
- Images demonstrating WRKY60, WRKY40, and WRKY18 localization per se should be added.
- Figure S1 – based on what author claims that GFP signal is localized to the autophagosomes and nucleus. Additional experiments are needed to confirm that. Usually, autophagosomes are visualized under stress conditions (as for example, is described in the book: Live Reference Work: CELL BIOLOGY, doi 10.1007/978-1-4614-7881-2_2-1, or in the book: Plant Proteostasis doi: 10.1007/978-1-4939-3759-2_13), thus the intracellular distribution of GFP-ATG8 is expected to change under stress.
- Materials and methods section lacks the description of 35S::PopP2-mCherry construct, the origin of GFP-StATG8 transgenic plants, 35S::GFP-ATG8 construct and the source of PopP2-HF (this construct is not introduced and explained in the text).
Minor points:
L. 92 “…ATG8 in Solanaceous crops, including Arabidopsis thaliana…” sounds strange as A. thaliana is not a member of this family
L 188-189 PopP2-HF should be introduced and explained at the first mentioning (L188). L 189 – in which plants? N.benthamiana?
Fig 4, Fig 5c, Fig 6 – scale bars should be added to the images
Fig 5 – molecular weight markers should be designated
Comments on the Quality of English Language
English should be improved
Author Response
Comments and Suggestions for Authors
Reviewer 1
The manuscript “Protein interaction of potato autophagy-related StATG8 family proteins with pathogen effector and WRKY transcription factor in the nucleus” promises to reveal some novel facts about Solanum tuberosum ATG8 proteins functioning. It contains interesting results and conclusions, however, several important controls are lacking.
àAuthor response: This study serves as a preliminary investigation into the nuclear role of autophagy, which remains largely unexplored. While it may not fully address all the reviewers' expectations, recent findings demonstrating that not only ATG8 proteins but also the autophagy-related protein ATG6 interact with the plant immune regulator NPR1 in the nucleus—and modulate their protein stability and degradation-highlight the nuclear involvement of autophagy components. These results support the relevance of our findings, in which ATG8 interacts with pathogen effectors and WRKY transcription factors, suggesting a broader regulatory role of autophagy in nuclear protein dynamics. The authors sincerely thank the reviewer for their insightful comments, which helped improve the quality of this manuscript.
1.Abstract should be re-written and made more accurate. It should contain an introduction sentence on PopP2 effector at least.
àAuthor response: The reviewer’s comment is well taken. It is agreed that the original abstract was overly concise and did not adequately convey the core findings and significance of the study. In response, the abstract has been thoroughly revised and expanded to provide a clearer and more detailed summary of the research, as shown below.
‘Autophagy is an essential eukaryotic catabolic process through which damaged or superfluous cellular components are degraded and recycled via the formation of double-membrane autophagosomes. In plants, autophagy-related genes (ATGs) are primarily expressed in the cytoplasm and are responsible for orchestrating distinct stages of autophagosome biogenesis. Among these, ATG8 proteins, orthologous to the mammalian LC3 family, are conserved ubiquitin-like modifiers that serve as central hubs in selective autophagy regulation. Although ATG8 proteins are localized in both the cytoplasm and nucleus, their functions within the nucleus remain largely undefined. In the present study, the ATG8-interacting motif (AIM) was identified and functionally characterized in the potato ATG8 homolog (StATG8), demonstrating its capacity for selective target recognition. StATG8 was shown to form both homodimeric and heterodimeric complexes with other ATG8 isoforms, implying a broader regulatory potential within the ATG8 family. Notably, StATG8 was found to interact with the Ralstonia solanacearum type III effector PopP2, a nuclear-localized acetyltransferase, suggesting a possible role in effector recognition within the nucleus. In addition, interactions between StATG8 and transcription factors AtWRKY40 and AtWRKY60 were detected in both cytoplasmic autophagosomes and the nuclear compartment. These observations provide novel insights into the noncanonical, nucleus-associated roles of plant ATG8 proteins. The nuclear interactions with pathogen effectors and transcriptional regulators suggest that ATG8 may function beyond autophagic degradation, contributing to the regulation of nuclear signaling and plant immunity. These findings offer a foundational basis for further investigation into the functional diversification of ATG8 in plant cellular compartments.’
- Introduction should be improved. The first paragraph looks separated from the other text, there is no some logical transition from it to the story of autophagy. As to the autophagy part, in my opinion, no need to compare animal and plant cells. Moreover, this part of the introduction doesn’t properly introduce the subject and the object and doesn’t help to understand the problem regarded in the paper. The Intro section lacks clarity and structure as well as some essential information on ATG8 functioning, AIM significance and ATG8 interactions with pathogens effectors. I would recommend to the author to focus on plant autophagy not going into animal and yeast autophagy in details. In the previous paper on potato ATG8 family the introduction section was much more informative and succinct.
àAuthor response: In accordance with the reviewer’s advice, the overly broad and general description of autophagy has been reduced. The main objective of this study is to suggest, although not conclusively, that autophagy-related proteins may function in the nucleus, based on their interactions with an effector protein and WRKY transcription factors. The Introduction section has been revised as much as possible to reflect the reviewer’s comments. In addition, a recent study reporting the nuclear role of another autophagy-related protein, ATG6, in regulating the immunity-related protein NPR1 has been cited (The changes were made on pages 3–4, lines 49–59) and incorporated into the reference list.
- I'm confused with the section 2.1. (Results) and don't understand its purpose in the Results. It looks out of place here as it discusses the previously obtained results on StATG8 and AtATG8 proteins (or it is written in a way that creates such an impression). Thus, it's better to move it to the Introduction.
àAuthor response: As suggested by the reviewer, a comprehensive investigation was conducted into previous studies and bioinformatic information regarding the evolutionary conservation of the StATG8 family and their known protein interactors, particularly those involved in cytoplasmic interactions. This analysis was intended to emphasize a key point of our study: although nuclear interactors were not clearly detected in the current IP-MS dataset, the experimental findings demonstrate that protein–protein interactions indeed occur within the nucleus.
In light of the reviewer’s comments, it is acknowledged that such content may not be ideal for inclusion in the Introduction, as it is based on external data analysis. Therefore, we kindly ask the reviewer to reconsider allowing this content to be presented in Figure 1, as it supports the main findings of this study and provides necessary context for the nuclear interactions revealed in our experiments.
L 124-126 contains a rather speculative statement on the mechanism underlying shuttling ability of ATG8
àAuthor response: New live-cell imaging data and 3D analysis using GFP-ATG8 have been included (Supplemental movie 1 and Figure 2). As pointed out by the reviewer, nucleocytoplasmic shuttling of ATG8 can be clearly observed in these images.
- I do not agree with the author that BiFC system could be used to confirm homodimerization. The described experimental setup that includes the same protein fused with split fluorophore is usually used to confirm that the system works properly and both parts of the fluorophore are produced. To make the result a proper basis for the conclusion made by the author an unambiguous negative control should be added to this experiment – for example some mutant variant of StATG8 that lacks a domain essential for interaction with partners. Such a control must be added to that experiment (Fig 3).
àAuthor response: As rightly pointed out by the reviewer, BiFC assays are prone to yielding false-positive or nonspecific signals, and negative interactions may often be observed. In response to this concern, yeast two-hybrid (Y2H) assays were conducted, and no interactions were detected in the negative control conditions. These results are presented alongside the BiFC data to provide complementary validation.
While the reviewer’s caution is well-founded, it is believed that the combination of these two independent experimental approaches offers a reliable confirmation of the observed protein–protein interactions.
- Which of StATG8 was used to generate GFP-StATG8 fuse used for experiments described in sections 2.3, 2.4 and for stably transformed A. thaliana plants?
àAuthor response: Transgenic Arabidopsis lines were generated for all members of the StATG8 family and were utilized to examine their subcellular localization in Figure3. Transient expression assays using N. benthamiana were conducted in Sections 2.3 and 2.4 to investigate BiFC assay.
- I wonder if there is some more accurate marker for autophagosomes, because in all figures where the author use GFP-StATG8 as a marker, the intracellular distribution is not very specific – GFP fluorescence is detected in the nucleus, cytoplasm and (not obvious for me and not convincing) autophagosomes. In addition, it is recommended to apply concavalin A to stabilize autophagosomes (Qi et al., 2023 doi 10.1007/s44307-023-00002-8)
àAuthor response: As the reviewer has pointed out, other chemical markers can be used to monitor autophagy. However, ATG8 remains the most reliable and widely accepted marker for autophagosomes. The ATG8-GFP used in this study is a commonly employed autophagy marker, and its application here is appropriate.
Furthermore, concerns regarding spectral overlap or interference between GFP and mCherry in the confocal microscopy data are unfounded, as these fluorophores have distinct excitation and emission spectra that do not overlap, supporting the validity of the presented data.
While additional experiments using chemical markers could be useful to visualize fixed autophagosome formation, the current results effectively demonstrate localization in the nucleus, cytoplasm, and autophagosomes, which adequately supports the conclusions of this study.
- PopP2-mCherry fuse could produce cytoplasmic fluorescence because of the fusion degradation. To confirm its integrity, the western-blot analysis should be performed that could show if there is some free mCherry or other degradation products present.
àAuthor response: The reviewer’s comment is valid and appreciated. Although Western blot analysis was not included in the current study, previous attempts to detect the cytoplasmic and nuclear distribution of PopP2-mCherry by Western blot revealed that the protein level in the cytoplasmic fraction was extremely low, making it difficult to obtain a clear signal. This suggests that PopP2-mCherry may be present only in very small amounts in the cytoplasm. Therefore, the cytoplasmic localization observed in this study is likely the result of the high sensitivity of confocal microscopy.
The same concern is connected with nucleocytoplasmic distribution of WRKY (Fig 6). First, there should be a control presenting WRKY fused to either part of split fluorescent protein and co-expressed: this combination is expected to give only nuclear fluorescence. Only in this case the conclusion about WRKY re-localization could be made. Second, western-blot analysis of integrity of protein fusions should be confirmed.
àAuthor response: As rightly noted by the reviewer, BiFC assays can sometimes produce false-positive signals in protein–protein interaction studies. However, in this study, yeast two-hybrid (Y2H) assays were also performed, and the results confirmed that the interactions observed are direct and specific. Therefore, the combination of BiFC and Y2H provides complementary and reliable evidence supporting the protein–protein interactions described. In addition, following the reviewer’s suggestion, the results of the negative control experiment using mRYNE-empty vector and mRYCE-AtWRKY40 have been included in Supplemental Data S6.
- L 266-267 It is written “…putative candidates with AIM among Arabidopsis transcription factors were analyzed…”. However, the paper by Song et al, 2022 claims that StATG8-WRKY interaction is independent of the ATG8 interacting motif (at least in the abstract). Thus the questions arise: 1) why the putative candidate transcription factors were selected based on the presence of AIM? 2) why the search was performed again as it had been already described in Song et al., 2022?
àAuthor response: The reviewer’s comment is appreciated and indeed valid. Bioinformatic analysis confirmed that putative AIM (ATG8-interacting motif) sequences are frequently found in transcription factors. This supports the possibility that ATG8 and other autophagy-related proteins may participate in protein–protein interactions within the nucleus, which is the main focus of the current study.
Previous work has demonstrated the interaction between StATG8 and WRKY transcription factors through co-immunoprecipitation (coIP). However, further investigation into the specificity of these interactions—such as mutational analysis of the AIM motif—is planned for future studies. Due to current limitations in funding and research resources, these additional experiments could not be conducted as part of this study. Nevertheless, in accordance with the reviewer’s suggestion, we fully acknowledge the importance of such analyses and intend to explore them in future research.
- Images demonstrating WRKY60, WRKY40, and WRKY18 localization per se should be added.
àAuthor response: WRKY18, WRKY40, and WRKY60 are well-characterized transcription factors that are known to localize to the nucleus. Therefore, additional experiments specifically addressing their subcellular localization were not conducted in response to the reviewer’s comment. The nuclear localization of these WRKY proteins has been clearly demonstrated in the literature, as referenced in the following study [1].
- Figure S1 – based on what author claims that GFP signal is localized to the autophagosomes and nucleus. Additional experiments are needed to confirm that. Usually, autophagosomes are visualized under stress conditions (as for example, is described in the book: Live Reference Work: CELL BIOLOGY, doi 10.1007/978-1-4614-7881-2_2-1, or in the book: Plant Proteostasis doi: 10.1007/978-1-4939-3759-2_13), thus the intracellular distribution of GFP-ATG8 is expected to change under stress.
àAuthor response: The reliability and widespread use of ATG8-GFP as a marker for autophagosomes in plants is well supported in the literature. For example, the following reference provides an overview of autophagy markers used in plant studies, including ATG8-GFP [2, 3].
- Materials and methods section lacks the description of 35S::PopP2-mCherry construct, the origin of GFP-StATG8 transgenic plants, 35S::GFP-ATG8 construct and the source of PopP2-HF (this construct is not introduced and explained in the text).
àAuthor response: In accordance with the reviewer’s suggestion, the relevant reference has been included in the revised manuscript [4, 5] .
Minor points:
- 92 “…ATG8 in Solanaceous crops, including Arabidopsis thaliana…” sounds strange as A. thaliana is not a member of this family
àAuthor response: The reviewer’s comment is valid; the original expression was somewhat inaccurate. The sentence has been revised as follows to improve clarity and precision:
‘Based on this, a phylogenetic analysis was conducted using ATG8 protein sequences from Solanaceae crops and Arabidopsis thaliana, revealing that they are grouped into two distinct clades.’
L 188-189 PopP2-HF should be introduced and explained at the first mentioning (L188). L 189 – in which plants? N.benthamiana?
àAuthor response: The reference has been added, and the experiment involves transient expression of Ralstonia PopP2-HF along with autophagy-related genes, ATG6 and ATG8, in N. benthamiana leaves.
Fig 4, Fig 5c, Fig 6 – scale bars should be added to the images
àAuthor response: As pointed out by the reviewer, the scale bar was displayed in a font size that was too small to be clearly visible. This has been corrected in the revised version.
Fig 5 – molecular weight markers should be designated
àAuthor response: As suggested by the reviewer, the necessary revision has been made.
References
- Xu X, Chen C, Fan B, Chen Z: Physical and functional interactions between pathogen-induced Arabidopsis WRKY18, WRKY40, and WRKY60 transcription factors. Plant Cell 2006, 18(5):1310-1326.
- Kumaran G, Pathak PK, Quandoh E, Devi J, Mursalimov S, Alkalai-Tuvia S, Leong JX, Schenstnyi K, Levin E, Üstün S et al: Autophagy restricts tomato fruit ripening via a general role in ethylene repression. New Phytol 2025, 246(6):2392-2404.
- Liu R, Zhang R, Yang Y, Liu X, Gong Q: Monitoring Autophagy in Rice With GFP-ATG8 Marker Lines. Front Plant Sci 2022, 13:866367.
- Sarris Panagiotis F, Duxbury Z, Huh Sung U, Ma Y, Segonzac C, Sklenar J, Derbyshire P, Cevik V, Rallapalli G, Saucet Simon B et al: A Plant Immune Receptor Detects Pathogen Effectors that Target WRKY Transcription Factors. Cell 2015, 161(5):1089-1100.
- Dagdas YF, Belhaj K, Maqbool A, Chaparro-Garcia A, Pandey P, Petre B, Tabassum N, Cruz-Mireles N, Hughes RK, Sklenar J et al: An effector of the Irish potato famine pathogen antagonizes a host autophagy cargo receptor. Elife 2016, 5.
- Xu X, Chen C, Fan B, Chen Z: Physical and functional interactions between pathogen-induced Arabidopsis WRKY18, WRKY40, and WRKY60 transcription factors. Plant Cell 2006, 18(5):1310-1326.
- Kumaran G, Pathak PK, Quandoh E, Devi J, Mursalimov S, Alkalai-Tuvia S, Leong JX, Schenstnyi K, Levin E, Üstün S et al: Autophagy restricts tomato fruit ripening via a general role in ethylene repression. New Phytol 2025, 246(6):2392-2404.
- Liu R, Zhang R, Yang Y, Liu X, Gong Q: Monitoring Autophagy in Rice With GFP-ATG8 Marker Lines. Front Plant Sci 2022, 13:866367.
- Sarris Panagiotis F, Duxbury Z, Huh Sung U, Ma Y, Segonzac C, Sklenar J, Derbyshire P, Cevik V, Rallapalli G, Saucet Simon B et al: A Plant Immune Receptor Detects Pathogen Effectors that Target WRKY Transcription Factors. Cell 2015, 161(5):1089-1100.
- Dagdas YF, Belhaj K, Maqbool A, Chaparro-Garcia A, Pandey P, Petre B, Tabassum N, Cruz-Mireles N, Hughes RK, Sklenar J et al: An effector of the Irish potato famine pathogen antagonizes a host autophagy cargo receptor. Elife 2016, 5.

Reviewer 2 Report
Comments and Suggestions for Authors
The research work is carried out well, and the experiments performed are appropriate for beginning to elucidate the role played by the proteins involved in the autophagosome.
The following comments and suggestions are intended to make the manuscript more attractive.
Table S1. Please enter the directionality of the primer oligonucleotides.
Define some of the abbreviations that appear for the first time in the manuscript, for example: IP-MS
line 107: Mention some examples of such organelles.
Fig. 1 . Name of microorganisms in italics.
Place the metric scale on the figures: 2, 3, 4, and 6.
Lines 397-399: Although the manuscript shows protein interactions using two-hybrid assays, it would be interesting if you included a diagram of some of the most relevant interactions, for example with the transcription factor, using this tool in the supplementary material. (This is only a suggestion.)
Lines 445-446. You need to describe the plasmid constructs in more detail.
Lines 448-451: Method reference.
Lines 453-456: Method reference.
Line 482: Perform the necessary calculations to put x g in place of rpm.
Author Response
Reviewer 2
The research work is carried out well, and the experiments performed are appropriate for beginning to elucidate the role played by the proteins involved in the autophagosome.
The following comments and suggestions are intended to make the manuscript more attractive.
ïƒ Author response: This study serves as a preliminary investigation into the nuclear role of autophagy, which remains largely unexplored. While it may not fully address all the reviewers' expectations, recent findings demonstrating that not only ATG8 proteins but also the autophagy-related protein ATG6 interact with the plant immune regulator NPR1 in the nucleus—and modulate their protein stability and degradation-highlight the nuclear involvement of autophagy components. These results support the relevance of our findings, in which ATG8 interacts with pathogen effectors and WRKY transcription factors, suggesting a broader regulatory role of autophagy in nuclear protein dynamics. The authors sincerely thank the reviewer for their insightful comments, which helped improve the quality of this manuscript.
Table S1. Please enter the directionality of the primer oligonucleotides.
ïƒ Author response: The manuscript has been revised in accordance with the reviewer’s comments.
Define some of the abbreviations that appear for the first time in the manuscript, for example: IP-MS
ïƒ Author response: All abbreviated terms have been written out in full, and the usage of acronyms has been rechecked throughout the manuscript. Thank you for the valuable suggestion.
line 107: Mention some examples of such organelles.
ïƒ Author response: The manuscript has been revised in accordance with the reviewer’s comments.
Fig. 1 . Name of microorganisms in italics.
ïƒ Author response: Thank you for pointing out the need to italicize Fungi. The correction has been made.
Place the metric scale on the figures: 2, 3, 4, and 6.
ïƒ Author response: The revision has been made according to your advice. Thank you for pointing out the inconsistent sizing-some parts were displayed too small while others were not, and your guidance helped correct this.
Lines 397-399: Although the manuscript shows protein interactions using two-hybrid assays, it would be interesting if you included a diagram of some of the most relevant interactions, for example with the transcription factor, using this tool in the supplementary material. (This is only a suggestion.)
ïƒ Author response: Thank you for your valuable comments. This study presents a preliminary investigation of nuclear-localized ATG8-interacting proteins that were not identified through IP-MS. As suggested, we plan to employ alternative experimental approaches to further identify these proteins. We also intend to improve the clarity of presentation, following your recommendation for a more accessible format.
Lines 445-446. You need to describe the plasmid constructs in more detail.
ïƒ Author response: The PopP2-HF and GFP-AtATG6 constructs were generously provided by collaborating researchers. This has been appropriately noted in the Materials and Methods section.
‘The PopP2-HF [4] and GFP-AtATG6 constructs were generously obtained from the laboratories of Prof. Jonathan Jones (The Sainsbury Laboratory, UK) and Dr. Yasin Dagdas (Gregor Mendel Institute of Molecular Plant Biology, Austria).’
Lines 448-451: Method reference.
ïƒ Author response: Author response: In response to the reviewer’s suggestion, relevant references have been added to the manuscript.
Lines 453-456: Method reference.
ïƒ Author response: In response to the reviewer’s suggestion, relevant references have been added to the manuscript.
Line 482: Perform the necessary calculations to put x g in place of rpm.
ïƒ Author response: The revision was made according to the reviewer's suggestion, and the speed was changed to 15,000 × g.
References
- Xu X, Chen C, Fan B, Chen Z: Physical and functional interactions between pathogen-induced Arabidopsis WRKY18, WRKY40, and WRKY60 transcription factors. Plant Cell 2006, 18(5):1310-1326.
- Kumaran G, Pathak PK, Quandoh E, Devi J, Mursalimov S, Alkalai-Tuvia S, Leong JX, Schenstnyi K, Levin E, Üstün S et al: Autophagy restricts tomato fruit ripening via a general role in ethylene repression. New Phytol 2025, 246(6):2392-2404.
- Liu R, Zhang R, Yang Y, Liu X, Gong Q: Monitoring Autophagy in Rice With GFP-ATG8 Marker Lines. Front Plant Sci 2022, 13:866367.
- Sarris Panagiotis F, Duxbury Z, Huh Sung U, Ma Y, Segonzac C, Sklenar J, Derbyshire P, Cevik V, Rallapalli G, Saucet Simon B et al: A Plant Immune Receptor Detects Pathogen Effectors that Target WRKY Transcription Factors. Cell 2015, 161(5):1089-1100.
- Dagdas YF, Belhaj K, Maqbool A, Chaparro-Garcia A, Pandey P, Petre B, Tabassum N, Cruz-Mireles N, Hughes RK, Sklenar J et al: An effector of the Irish potato famine pathogen antagonizes a host autophagy cargo receptor. Elife 2016, 5.

Round 2
Reviewer 1 Report
Comments and Suggestions for Authors
The author addressed most of my comments in the reply and modified the manuscript. Abstract and the Introduction section have been improved and look much better now.
Unfortunately, author couldn’t convince me about the section 2.1. (Results). The phylogenetic tree has already been constructed and presented in Song et al., 2022. Thus, I do not see any novelty in this result even though ATG8 from Glycine max (why this species?). Moreover, “Potato StATG8-2.1 and StATG8-2.2 exhibited differences in their DNA sequences, but their amino acid sequences were completely identical (Fig. 1a)” – has been also demonstrated by Song et al. Lines 94-99 contain information on AtATG8 but I don’t understand its purpose in this MS. And L. 100-111 contain a review/discussion of the results obtained by Zess et al., 2019. In my opinion, the section 2.1. couldn’t be included in the Results. It could be revised and used for the Introduction or Discussion section.
As to my concern about the results obtained using BiFC system, the author added correct negative control making the results more convincing.
Does the combination of mRYNE/mRYCE (without any fused proteins) give fluorescence in this experimental system?
Why in case of PopP2 different fusions – N- or C-terminal – are used? for BiFC it is mRYNE-PopP2, while with full mCherry – vice versa? Actually, it could affect stability and integrity of the fusion protein.
L. 257 – PopP2-HF should be introduced. Although in M&M section the reference was added, the abbreviation HF should be explained.
Materials and methods section still lacks the description of 35S::PopP2-mCherry construct as well as mRYNE-PopP2 construct.
I noticed that the title of the manuscript contains repetition of the word "protein". The author could regard changing the title as follows:
"Interaction of potato autophagy-related StATG8 family
proteins with pathogen effector and WRKY transcription factor
in the nucleus"
Some phrases should be checked and corrected
Author Response
Comments and Suggestions for Authors
Author's Reply to the Review Report (Reviewer 1)
The author addressed most of my comments in the reply and modified the manuscript. Abstract and the Introduction section have been improved and look much better now.
àAuthor response: Thank you once again for your thoughtful and constructive feedback. I fully acknowledge that there are still many aspects of the manuscript that could be improved. While I would very much like to explore additional experiments as suggested, current limitations in research funding and resources make it challenging to carry out more extensive analyses. Nonetheless, I truly appreciate your invaluable suggestions, which have helped improve the clarity and direction of this study. Although this work is modest, I hope it can serve as a meaningful reference in understanding how autophagy-related proteins such as ATG8 may shuttle within the nucleus and potentially contribute to cellular homeostasis through yet-to-be-elucidated regulatory mechanisms.
Unfortunately, author couldn’t convince me about the section 2.1. (Results). The phylogenetic tree has already been constructed and presented in Song et al., 2022. Thus, I do not see any novelty in this result even though ATG8 from Glycine max (why this species?). Moreover, “Potato StATG8-2.1 and StATG8-2.2 exhibited differences in their DNA sequences, but their amino acid sequences were completely identical (Fig. 1a)” – has been also demonstrated by Song et al. Lines 94-99 contain information on AtATG8 but I don’t understand its purpose in this MS. And L. 100-111 contain a review/discussion of the results obtained by Zess et al., 2019. In my opinion, the section 2.1. couldn’t be included in the Results. It could be revised and used for the Introduction or Discussion section.
àAuthor response: We sincerely appreciate the reviewer’s detailed comments on Results section 2.1 and the important point regarding overlap with previously published work. Upon careful reconsideration, we agree that it is appropriate to revise the manuscript accordingly, and we have relocated the relevant content to the Introduction section as suggested. We thank the reviewer once again for the valuable advice, which has greatly improved the clarity and organization of the manuscript.
As to my concern about the results obtained using BiFC system, the author added correct negative control making the results more convincing.
àAuthor response: As the reviewer pointed out, BiFC serves as one method to verify protein–protein interactions. Therefore, complementary approaches such as yeast two-hybrid (Y2H) or GST pull-down assays can be employed to further validate these interactions. In this study, however, the PopP2–StATG8 interaction has already been confirmed in vivo by co-immunoprecipitation (coIP). Additionally, the direct transfer of PopP2 acetylation activity to StATG8 protein was demonstrated by the acetylation assay shown in Figures 5d and 5e. These complementary experiments provide strong evidence supporting the protein interaction described.
Does the combination of mRYNE/mRYCE (without any fused proteins) give fluorescence in this experimental system?
àAuthor response: The mRYNE/mRYCE empty vectors used as negative controls did not produce any BiFC signals in our experiments. This observation is consistent with previous reports, and the relevant reference has been included in the revised manuscript.
Why in case of PopP2 different fusions – N- or C-terminal – are used? for BiFC it is mRYNE-PopP2, while with full mCherry – vice versa? Actually, it could affect stability and integrity of the fusion protein.
àAuthor response: I appreciate the reviewer’s insightful suggestion regarding the fusion position of the BiFC tags. In my study, the BiFC system employed involved fusing both the N- and C-terminal fragments of mCherry to the N-terminus of PopP2. While it is indeed possible to fuse the BiFC tag at the C-terminus, PopP2 is a relatively stable protein and does not present issues for generating a BiFC signal with the current fusion strategy. If there had been a problem with the fusion position interfering with protein interaction, PopP2 would not have shown any interaction with ATG8 in the first place. Therefore, I believe the chosen approach is appropriate for detecting the PopP2–ATG8 interaction in this study.
- 257 – PopP2-HF should be introduced. Although in M&M section the reference was added, the abbreviation HF should be explained.
àAuthor response: Apologies for the lack of clarity in the previous response. The abbreviation “HF” was used without proper explanation, under the assumption that it would be generally recognized. In the revised manuscript, the term has now been clearly defined as a dual affinity tag consisting of 6×His and Flag epitopes (His₆-Flag), and this clarification has been incorporated into the Materials and Methods section accordingly.
‘The PopP2 gene was cloned into a plasmid vector harboring a C-terminal HF (His₆-Flag) tag [31]’
Materials and methods section still lacks the description of 35S::PopP2-mCherry construct as well as mRYNE-PopP2 construct.
àAuthor response: I apologize for the insufficient explanation in my previous response. To clarify, the PopP2-mCherry construct, which has been previously described in the literature [57], was kindly provided by Prof. Jonathan Jones (The Sainsbury Laboratory, UK). Additionally, I generated the mRYNE-PopP2 BiFC construct using conventional restriction enzyme-based cloning methods. The primer sequences used for this cloning are provided in Table S1 of the revised manuscript. I hope this clarification addresses your concerns.
I noticed that the title of the manuscript contains repetition of the word "protein". The author could regard changing the title as follows:
"Interaction of potato autophagy-related StATG8 family
proteins with pathogen effector and WRKY transcription factor
in the nucleus"
àAuthor response: Thank you for your valuable suggestion regarding the title. I agree that avoiding redundant terms in the title enhances clarity and readability. In accordance with your recommendation, the title has been revised as follows:
‘Interaction of potato autophagy-related StATG8 family proteins with pathogen effector and WRKY transcription factor in the nucleus.’
